# Acquisition of Chinese Verb Separation by Adult L2 Learners

**Zhe Gao** [1,*], **Seth Wiener** [1]  **and Brian MacWhinney** [1,2]

1 Department of Modern Languages, Carnegie Mellon University, Pittsburgh, PA 15213, USA
2 Department of Psychology, Carnegie Mellon University, Pittsburgh, PA 15213, USA
* Correspondence: zheg@andrew.cmu.edu

**Abstract:** Chinese verb separation is a grammatical construction in which a bimorphemic verb compound separates and expands into a verb phrase. For example, to say "sang one song", one must insert the aspect marker *le* and classifier phrase *yī shǒu* between the verb *chàng gē* 'sing-song' as in *chàng le yī shǒu gē* (sing-LE-one-classifier-song). Adult second language (L2) learners face at least three problems related to acquiring verb separation: separation patterns are complex and numerous, classroom oral input is limited, and L1 transfer often fails. To better understand L2 learners' acquisition of verb separation, this study tested 28 adult L2-Chinese classroom learners. Three tasks were administered online: verb decomposition, grammaticality judgment, and oral translation. On average, only 37% of verbs were accurately decomposed, 63% of verbs were accurately judged to be grammatical, and 19% of verbs were orally produced with the correct separation pattern. Chinese verb separation, particularly decomposition and oral production, is thus extremely challenging for L2 learners to acquire—even for advanced learners with a relatively large vocabulary size. The results are discussed in terms of the Unified Competition Model and L2 Chinese pedagogy.

**Keywords:** Chinese verb compounds; separation; online experiment; L2 learning

## 1. Introduction

Verbs are basic building blocks for sentences. They connect noun phrases by assignment to case roles, and verb phases can further specify aspect, tense, mood, and agreement. Mandarin Chinese (hereafter 'Chinese') verbs consist of one to four morphemes but function as a single word (Chao 1968; Li and Thompson 1981). The present study focuses on bimorphemic Chinese verbs, specifically those with verb–object/noun (VO), verb–verb (VV) and verb–resultative (VR) structures. Together, these three forms constitute 83% of the bimorphemic verb compounds in Chinese (Chen and Duanmu 2016). Even though these verbs contain only two morphemes, their separability may pose challenges for L2 Chinese learners with a European language as their first language (L1) for at least three reasons.

First, Chinese verb separation is a complex system. Separable verb compounds can be expanded through insertion, duplication and reordering of parts of the verb and additional elements. Consider the construction of insertion as an example. In (1), the aspect marker ZHE[1] is inserted between the VO verb 唱歌 (*chàng gē*) to indicate a progressive action.

(1)        他              唱            着            歌。
              *tā*            *chàng*        *zhe*         *gē*
              he            sing-         ZHE-        song
              'He is singing.'

Different categories of verbs allow for different separation patterns. VV verbs such as 学习 (*xué xí*, study–study, 'to study') can never be separated by the ZHE progressive aspect marker. The interactions between the morphological structures of verbs and the separation constructions generate a total of 11 verb separation patterns. Section 2 outlines these 11 patterns in detail following a traditional approach (Chao 1968; Li and Thompson 1981; Packard 2000).

Second, classroom L2 learners are often limited in their information to facilitate learning verb separation. Through searching various Chinese corpora, Wang (2008) and Siewierska et al. (2010) have revealed that verb separation primarily exists in spoken Chinese. For example, Wang (2008) examined a corpus of 115 million characters across various genres/types of Chinese. The author found that VO verbs separated 13 times more frequently in speech than in written language. Classroom learners, particularly those who meet for only a few days a week, are thus typically limited in their exposure to verb separation patterns. Moreover, because L2 textbooks are often concise introductions to the language, popular textbooks for L2 Chinese learners (e.g., Wu et al. 2011; Wu and Yu 2012) do not provide extensive linguistic analyses for learners to fully understand all 11 separation patterns. Similarly, a bimorphemic verb's morpheme meanings are rarely taught systematically, and compounds are often taught as units (e.g., He et al. 2008; Liu 2007).

Third, the concept of verb separation may seem quite foreign to speakers of languages that make no use of this type of structure. Cheng and Sybesma (1998) have observed that many Chinese VO verbs have monomorphemic English equivalents which, by definition, are not separable. See the examples in (2).

(2)   a.   跑步     |   b.   开 车     |   c.   搬 家     |   d.   走 路

       *pǎo bù*     |      *kāi chē*     |      *bān jiā*     |      *zǒu lù*

       run-step     |      drive-car     |      move-home     |      walk-road

       'run'     |      'drive'     |      'move'     |      'walk'

Controlling verb separation therefore involves two steps: decomposing bimorphemic compounds into individual morphemes and fitting the morphemes into separation patterns. Studies have shown that decomposition is necessary in the processing of multimorphemic Chinese words (Tsang and Chen 2013a, 2013b, 2014; Wu et al. 2017; Zhou and Marslen-Wilson 1995). For example, using two masked priming tasks, Tsang and Chen (2014) found that native Chinese speakers quickly activated the meanings of the constituent morphemes of a compound, even for opaque compounds, i.e., even if the compound meaning has little to no overlap with the joint meanings of its constituent morphemes. For example, in (3), opaque (3a) and transparent compounds (3b) similarly facilitated the recognition of the target compound (3c).

(3)   a.   雷       达     |   b.   雷       雨     |   c.   闪       电

       *léi*       *dá*     |      *léi*       *yǔ*     |      *shǎn*       *diàn*

       thunder-reach     |      thunder-rain     |      flash-electric

       'radar'     |      'thunderstorm'     |      'lightening'

Can L2 Chinese learners also acquire the morpho-semantic and orthographic cues needed for decomposing multimorphemic words? Chen et al. (2020) found that intermediate and above L2-Chinese learners were able to decompose transparent Chinese compound nouns such as 车票 (*chē piào*, car-ticket, 'bus ticket') but not opaque nouns such as 冷门 (*lěng mén*, cold-door, 'unpopular') in two offline translation tasks. Yet, even if learners acquire the skills of decomposing verb compounds, they need to also learn the syntactic and morphological cues determining the application of specific verb separation patterns. It remains unclear how and when adult L2 learners develop control of the system of verb separation. The current study examines adult L2 learners' knowledge on verb separation to pinpoint areas of learning difficulties.

## 2. Grammatical Analysis of Chinese Verb Separation

Following a traditional approach (Chao 1968; Li and Thompson 1981; Packard 2000), the current study categorizes Chinese bimorphemic verb compounds according to the part of speech of their constituent morphemes and the semantic relationship between the morphemes. VO verbs are composed of a verb and an object of that verb (e.g., (4a)); VR verbs are composed of a verb indicating action and a resultative component indicating the results of the action (e.g., (4b)). VV verbs refer to those with a coordinate structure such as 运动 in (4c). Table 1 summarizes the 11 separation patterns. These were observed

by Chao (1968) and Li and Thompson (1981) and were also found in Chinese corpora by Wang (2008) and Siewierska et al. (2010). Meanwhile, they appear in Chinese textbooks for L2-Chinese learners.

(4)  a.  唱　　　　歌　　　　|  b.  降　　　　低　　　　|  c.  运　　　　动
　　　*chàng*　　*gē*　　|　　*jiàng*　　*dī*　　|　　*yùn*　　　*dòng*
　　　sing -　　song　　|　　lower-low　　|　　move-move
　　　'sing'　　　　　　|　　'low'　　　　|　　'exercise'

**Table 1.** Summary of verb separation.

| Construction | Insertion | | Post-Repetition | Pre-Repetition | Reduplication | Topicalization |
|---|---|---|---|---|---|---|
| Cues | Aspect Markers, Operators | DE, BU | Operators | A-NOT-A | Delimitative | Topic |
| VO-verb | + | - | + | + | + | + |
| VV-verb | - | - | - | + | + | - |
| VR-verb | - | + | - | + | - | - |
| Alternative (non-) separation patterns | +/- | + | + | + | + | + |
| Total number of separation patterns | 6 | | 2 | 1 | 1 | 1 |

Note: +: The verb can undergo separation when used with this construction. -: The verbs do not need separation to fulfill a certain function.

The first major grammatical construction, insertion, has two forms or six separation patterns in total. VO verbs permit insertion between the two morphemes for the aspect markers LE (for perfective aspect: 5a), ZHE (for progressive aspect: 5b) and GUO (for experiential aspect: 5c) (Li and Thompson 1981), as well as certain operators such as the adverbial phrases composed of numbers, classifiers (CLs) and nouns in (5d) (Chao 1968). One exception is that LE can be inserted into a VO verb when marking perfective aspect (5a) or following the VO verb when marking inchoative aspect (5e). Siewierska et al. (2010) observed the insertion of wh-words in a VO verb such as (5f). This pattern was excluded from the current analysis because this pattern scarcely appeared in L2 learners' textbook materials.

(5)  a.  苏　　　轼　　　睡　　　了　　　觉。
　　　*sū*　　*shì*　　*shuì*　　*le*　　*jiào*
　　　Su　　Shi　　sleep (V)　　LE　　sleep (O)
　　　'Su Shi slept.'

　　　b.  苏　　　轼　　　睡　　　着　　　觉。
　　　*sū*　　*shì*　　*shuì*　　*zhe*　　*jiào*
　　　Su　　Shi　　sleep (V)　　ZHE　　sleep (O)
　　　'Su Shi is sleeping.'

　　　c.  苏　　　轼　　　睡　　　过　　　觉。
　　　*sū*　　*shì*　　*shuì*　　*guò*　　*jiào*
　　　Su　　Shi　　Sleep (V)　　GUO　　sleep (O)
　　　'Su Shi has slept.'

　　　d.  苏　　　轼　　　睡　　　了　　　八　　　个　　　小　　　时　　　的　　　觉。
　　　*sū*　　*shì*　　*shuì*　　*le*　　*bā*　　*gè*　　*xiǎo*　　*shí*　　*de*　　*jiào*
　　　Su　　Shi　　Sleep (V)　　LE　　eight　　CL　　hour　　　　DE　　sleep (O)
　　　'Su Shi slept for eight hours.'

　　　e.  苏　　　轼　　　睡　　　觉　　　了。
　　　*sū*　　*shì*　　*shuì*　　*jiào*　　*le*
　　　Su　　Shi　　sleep (V)　　sleep (O)　　LE
　　　'Su Shi slept.'

|   | f. | 你 | 担 | 什 | 么 | 心！ |
|---|---|---|---|---|---|---|
|   |   | *nǐ* | *dān* | *shén* | *me* | *xīn* |
|   |   | you | hold | WHAT |   | heart |

'You don't need to be worried!'

The second separation pattern of the insertion construction requires separation for VR verbs only for the modal DE ('be able to': 6a) and negation marker BU ('not be able to': 6b) (Li and Thompson 1981). As shown in (6c) and (6d), the separation pattern has a non-separation counterpart, the modal verb 能 (*néng*, be able to).

| (6) | a. | 我 | 拉 | 得 | 开 | 门。 |
|---|---|---|---|---|---|---|
|   |   | *wǒ* | *lā* | *de* | *kāi* | *mén* |
|   |   | I | pull | DE | opened | door |

'I am able to pull the door open.'

|   | b. | 我 | 拉 | 不 | 开 | 门。 |
|---|---|---|---|---|---|---|
|   |   | *wǒ* | *lā* | *bù* | *kāi* | *mén* |
|   |   | I | pull | BU | opened | door |

'I am not able to pull the door open.'

|   | c. | 我 | 能 | 拉 | 开 | 门。 |
|---|---|---|---|---|---|---|
|   |   | *wǒ* | *néng* | *lā* | *kāi* | *mén* |
|   |   | I | be able to | pull | opened | door |

'I am able to pull the door open.'

|   | d. | 我 | 不 | 能 | 拉 | 开 | 门。 |
|---|---|---|---|---|---|---|---|
|   |   | *wǒ* | *bù* | *néng* | *lā* | *kāi* | *mén* |
|   |   | I | not | be able to | pull | opened | the door |

'I am not able to pull the door open.'

The second grammatical construction is post-repetition. The verb morpheme of a VO verb is repeated and placed after the whole verb compound but before the operators to convey duration (7a), frequency (7b) and status (7c). Note that the insertion pattern like (5d) and the post-repetition patterns of (7a) and (7b) are interchangeable. In other words, a VO verb compound can undergo either insertion or post-repetition with an adverb for duration or for frequency. Learners need to choose either of the patterns.

| (7) | a. | 我 | 睡 | 觉 | 睡 | 了 | 三 | 个 | 小 | 时。 |
|---|---|---|---|---|---|---|---|---|---|---|
|   |   | *wǒ* | *shuì* | *jiào* | *shuì* | *le* | *sān* | *gè* | *xiǎo* | *shí* |
|   |   | I | sleep (V) | sleep (O) | sleep (V) | LE | three | CL | hour |   |

'I have slept for three hours.'

|   | b. | 我 | 洗 | 澡 | 洗 | 了 | 两 | 次。 |
|---|---|---|---|---|---|---|---|---|
|   |   | *wǒ* | *xǐ* | *zǎo* | *xǐ* | *le* | *liǎng* | *cì* |
|   |   | I | wash | bath | wash | LE | two | time |

'I have bathed for two times.'

|   | c. | 他 | 跑 | 步 | 跑 | 得 | 很 | 快。 |
|---|---|---|---|---|---|---|---|---|
|   |   | *tā* | *pǎo* | *bù* | *pǎo* | *de* | *hěn* | *kuài* |
|   |   | he | run | step | run | DE | very | fast |

'He runs very fast.'

The third construction is pre-repetition for yes–no questions, which applies to all three verb types: VO (8a), VV (8b) and VR (8c). Chao (1968) observed that this repetition pattern has at least four varieties such as reduplicating the whole verb, and that some verbs were exceptional for certain varieties. However, the current study focuses on the variety that applies most widely, i.e., the reduplication of the first morpheme of the verb. This separation pattern also has a non-separation counterpart: 吗 (*ma*) as a sentence final particle.

(8)  a.  | 你 | 睡 | 不 | 睡 | 觉? |
|---|---|---|---|---|
| *nǐ* | *shuì* | *bù* | *shuì* | *jiào* |
| You | sleep (V) | NOT | sleep (V) | sleep (O) |

'Do you have a sleep?'

b.  | 你 | 喜 | 不 | 喜 | 欢 | 苹 | 果? |
|---|---|---|---|---|---|---|
| *nǐ* | *xǐ* | *bù* | *xǐ* | *huān* | *píng* | *guǒ* |
| You | like | NOT | like | happy | apple | |

'Do you like apples?'

c.  | 你 | 回 | 不 | 回 | 去? |
|---|---|---|---|---|
| *nǐ* | *huí* | *bù* | *huí* | *qù* |
| you | return | NOT | return | go? |

'Will you go back?'

The fourth grammatical construction is reduplication, meaning *a little bit* (Li and Thompson 1981). For VO verbs, only the first morpheme is reduplicated, as shown in Example (9a), while for VV verbs, the whole verb is reduplicated, as shown in (9b). Because reduplication signals actions, it is not applicable for stative verbs such as 喜欢 (*xǐ huān*, 'to like') and VR verbs indicating results. This separation pattern also has a non-separation counterpart: 一下 (*yī xià*) in (9c).

(9)  a.  | 他 | 洗 | 洗 | 澡， | 就 | 去 | 工 | 作 | 了。 |
|---|---|---|---|---|---|---|---|---|
| *tā* | *xǐ* | *xǐ* | *zǎo* | *jiù* | *qù* | *gōng* | *zuò* | *le* |
| He | wash | wash | bath | then | go | work | | LE |

'He took a quick shower and then went to work.'

b.  | 他 | 要 | 学 | 习 | 学 | 习 | 这 | 篇 | 论 | 文。 |
|---|---|---|---|---|---|---|---|---|---|
| *tā* | *yào* | *xué* | *xí* | *xué* | *xí* | *zhè* | *piān* | *lùn* | *wén* |
| he | want | study-study | | study-study | | this | CL | paper | |

'He wanted to study this paper a little bit.'

c.  | 他 | 要 | 学 | 习 | 一 | 下 | 这 | 篇 | 论 | 文。 |
|---|---|---|---|---|---|---|---|---|---|
| *Tā* | *yào* | *xué* | *xí* | *yī* | *xià* | *zhè* | *piān* | *lùn* | *wén* |
| He | want | study | study | a little bit | | this | CL | paper | |

'He wanted to study this paper a little bit.'

The fifth and final grammatical construction examined in this study is topicalization. This involves placing the object morpheme of a VO verb at the beginning of the sentence (Chao 1968; Packard 2000). In the example of (10) with the verb 唱歌 (*chàng gē*, sing-song, 'sing'), the object morpheme 歌 ('song') is placed at the beginning of the sentence. There are at least three alternative (non-)separation patterns for topicalization, including the canonical sentence structure, insertion and post-repetition.

(10)  | 歌 | 我 | 们 | 唱 | 了 | 一 | 个 | 小 | 时。 |
|---|---|---|---|---|---|---|---|---|
| *gē* | *wǒ* | *men* | *chàng* | *le* | *yī* | *gè* | *xiǎo* | *shí* |
| song | we | | sing | LE | one | CL | hour | |

'We sang for one hour.'

Importantly, the boundaries between two-morpheme compounds and phrases in Chinese are not always clear-cut (Feng 2001; Packard 2000; Yeh 2020). Some scholars treat all VO combinations as verb-object phrases (e.g., Wang 1946), while others (e.g., Duanmu 2016; Feng 2001; Huang et al. 2017; Siewierska et al. 2010) argue that separable VO verbs function as compounds in sentences with canonical word order, such as (11a), but as verb phrases in sentences with separation, such as (11b). This latter analysis fits in well with the current study because it aligns with the fact that the learner must acquire these forms as compounds while also learning to treat them as separable. The status of VV and VR verbs are usually excluded in the discussion of separable verbs, despite the fact that they undergo separation in the constructions with pre-repetition and insertion.

(11)  a.  

| 他 | 不 | 担 | 心 | 小 | 明。 |
|---|---|---|---|---|---|
| *tā* | *bù* | *dān* | *xīn* | *xiǎo* | *míng* |
| He | NOT | hold-heart | | Xiaoming | |

'He did not worry about Xiaoming.'

b.  

| 他 | 为 | 这 | 件 | 事 | 担 | 了 | 一 | 百 | 个 | 心。 |
|---|---|---|---|---|---|---|---|---|---|---|
| *tā* | *wèi* | *zhè* | *jiàn* | *shì* | *dān* | *le* | *yī* | *bǎi* | *gè* | *xīn* |
| He | for | this | CL | issue | hold | LE | one | hundred | CL | heart |

'He worried about this issue so much.'

To comprehensively investigate the challenges of learning verb separation from the L2-Chinese learner's perspective, the current study adopts a broad definition of compounds. Verb compounds were included in the study as long as they underwent one of the aforementioned separation patterns introduced in this section. Future studies will need to explore the topic of L2 learners' knowledge of compound-phrase boundaries, which is beyond the scope of the current study.

In summary, verb separation patterns depend on the morphological structure of the target verbs as well as the target grammatical construction. VO verbs have the most complex separation patterns, followed by VR verbs. VV verbs have the least separation patterns. Thus, we predict that VO verb separation patterns are the most difficult for L2-Chinese learners, followed by VR verb separation patterns. VV verb separation pattern is the easiest. Lastly, all the separation patterns have at least one alternative (non-)separation patterns except for the insertion pattern with the three aspect markers.

## 3. Theoretical Framework

The current study uses the Unified Competition Model as its theoretical framework (MacWhinney 1987, 1992, 1997a, 2012, 2018, 2021). The model posits that competition arises when L2 learners try to map multiple forms to one function or meaning. For L2 acquisition of verb separation, competition can sometimes occur when a separation pattern competes with its counterpart that does not separate the compound verb. For example, the sentence using the pre-repetition separation pattern in (12a) is equivalent to the sentence in (12b) using a sentence final particle 吗 (*ma*) for the yes–no question. Moreover, as previously noted, a verb compound can have more than one separation patterns. When discussing length of sleep, both insertion (5d) and post-repetition (7a) patterns convey the same meaning. The competition can cause learning and processing difficulties because the learners try to choose the "right" form to map the function of a sentence.

(12)  a.  

| 你 | 昨 | 天 | 跳 | 没 | 跳 | 舞? |
|---|---|---|---|---|---|---|
| *nǐ* | *zuó* | *tiān* | *tiào* | *méi* | *tiào* | *wǔ?* |
| You | yesterday | | jump | -NOT | -jump | -dance(N) |

'Did you dance yesterday?'

b.  

| 你 | 昨 | 天 | 跳 | 舞 | 了 | 吗? |
|---|---|---|---|---|---|---|
| *nǐ* | *zuó* | *tiān* | *tiào* | *wǔ* | *le* | *ma?* |
| You | yesterday | | jump | -dance | LE | MA |

'Did you dance yesterday?'

The second competition involves L1 transfer. L2 learners initially try to transfer everything from their L1 to the L2 (MacWhinney 2012). Transfer serves as a general approach that learners try, but competition arises when the L1 and L2 are vastly different, and learners need to construct their L2s. VO verbs can be mapped to similar verb–object phrases in European languages, as in the mapping of 唱歌 (*chàng gē*, sing-song, 'sing') to "sing a song" in English. However, as example (2) shows, many Chinese VO verbs have only mono-morphemic English equivalents, which, by definition, are not separable (Cheng and Sybesma 1998). On the syntactic level, the insertion pattern for VO verbs can be mapped with complex verb phrases in European languages. For example, 我做了一个梦 (*wǒ zuò le yī gè mèng*, I make LE one-CL dream) can be 'I dreamed a dream'. However, other separation patterns such as reduplication and post-repetition with adverbs are ungrammatical in English. The verb of an English VO phrase can never be reduplicated.

Consequently, L1 transfer does not work in many cases, and it does not help with many separation patterns. Thus, the novelty and complexity of Chinese separation patterns makes it difficult for L2 learners to find any clear mapping to parallels in L1-European languages.

To resolve these competitions and mapping problems, the Unified Competition Model holds that learning will proceed best if the relevant linguistic constructs, i.e., cues, are simple, highly valid, and frequently available (MacWhinney et al. 1984; MacWhinney 2012, 2021). Chinese verb separation is composed of two sets of cues: the morphological structures of the verb compounds, and the separation patterns. If explicit instruction is provided, it should be maximally simple and direct (MacWhinney 1997b). Unfortunately, for L2 learners, the morphological and syntactic cues of Chinese verb separation are often not fully available due to limited exposure to natural speech and non-explicit and unsystematic classroom instruction. Consequently, learners may struggle in learning verb separation.

## 4. Current Study

The current study aims to evaluate the knowledge of L2 Chinese learners with L1-European languages and locate the cues and skills which are especially challenging for them. Our three research questions (RQs) are as follows:

RQ1: What do L2 learners know about the decomposition of verb compounds?

RQ2: How do L2 learners process and produce the sentences containing verb separation patterns?

RQ3: What differences might there be in performance on different verb types?

## 5. Methods

### 5.1. Participants

In total, 30 L2-Chinese speakers (19 female, 10 male and 1 non-binary; mean age 20.53 years; SD = 3.35) were recruited from eight US universities. All participants dominantly used simplified Chinese orthography and had studied at least one semester in a classroom setting when participating in this study (mean length of Chinese learning was 4.87 years). Twenty-nine were native speakers of English, and one was Spanish–English bilingual. No participant spoke an Eastern Asian language such as Japanese or Korean as their L1 or L2. Following Montrul (2004) and Valdés (2000, p. 1), we defined heritage learners of Chinese as bilinguals whose home language is a Chinese language, and whose community language is not. Therefore, the study contained no heritage learners.

### 5.2. Materials

The target verbs for this study included 198 total unique verbs. Ideally, the target verbs and the vocabulary for sentences should be selected such that all the participants were familiar with the words. Realistically, elementary Chinese vocabulary was far too limited for stimuli creation. The target verbs were selected from the word lists of the standardized Chinese proficiency test HSK (Office of Chinese Language Council International 2007) and the vocabulary lists of Chinese textbooks Chinese Link (Elementary: Wu et al. 2011; Intermediate: Wu and Yu 2012). If the verbs were not found from the lists, the characters from those lists were used to generate the target verb compounds. All the target verbs were checked by five native speakers of Mandarin Chinese to ensure they were familiar. Materials, including the frequency of all verbs tested, are available on the OSF platform (https://osf.io/s3u2z/, accessed on 21 August 2022).

The experiment was composed of three tasks: a speeded verb decomposition task, a grammaticality judgement task, and an oral translation task. Two individual differences measures were also collected using the Language History Questionnaire 3 (LHQ 3) (Li et al. 2020) and L2 Chinese proficiency LEXTALE_CH task (Chan and Chang 2018).

### 5.2.1. Language History Questionnaire (LHQ3)

LHQ3 (Li et al. 2020) was used to collect the background information of the participants. In total, twenty-seven questions were included in this questionnaire. Besides the questions on the basic information such as age and gender, this questionnaire collected detailed information about the participants' language and culture background. For instance, the frequency of using each of their languages, their dialects, etc. Most importantly, the questionnaire calculated the aggregated scores for self-assessed global L2 proficiency. This score ranged from 0 to 1, with higher scores indicating higher Chinese proficiency. The questionnaire was implemented on its own platform (https://lhq-blclab.org/ accessed on 21 August 2022).

### 5.2.2. LEXTALE_CH

LEXTALE_CH measured L2 Chinese learners' proficiency by estimating their Mandarin vocabulary size (Chan and Chang 2018). This test contained 90 total items with 60 real Chinese characters and 30 nonce characters. These 90 items were presented to the participants in a random order. The participants were instructed to judge whether each character was a real or a nonce Chinese character via button press. Following Chan and Chang (2018), corrected accuracy (hits − 2 times false alarms) was used as it accounted for both correct hits and incorrectly accepted false alarms. The corrected accuracy ranged from −30 to 60. Higher score meant higher Chinese proficiency. LEXTALE_CH was administered on Qualtrics (2021).

### 5.2.3. Speeded Verb Decomposition Task

This task examined the participants' ability to decompose bimorphemic verb compounds. The participants were presented with the verbs in characters along with Pinyin and audio recordings, which were played automatically. Participants needed to indicate whether the translations of the verbs and their constituent characters were correct or not via button press. The maximum judgment time was five seconds for each verb and character. The constituent characters were presented immediately after each of the verbs.

The task contained 66 verbs, i.e., 22 verbs for VO, VR and VV structures, respectively. Each verb had 3 trials, i.e., 198 trials in total, the translation of the verb compound and 1 translation for each of the two constituent characters. Half of the verbs as well as their constituent characters had correct translations, while the other half had incorrect but semantically relevant translations. For example, in the trial of "看见 (*kàn jiàn*, to see)", the incorrect translation was "to hear". Following the principle of a counter-balanced design, two versions (Version A and Version B) were prepared. The verbs and morphemes with incorrect translations in Version A had correct translations in Version B, and vice versa.

A verb was considered to be accurately decomposed if the judgments regarding its meaning and the meanings of both its constituent characters were correct. That is, all three translation judgments had to be correct.

### 5.2.4. Grammaticality Judgement Task

This task examined whether the participants could accurately process the sentences with verb separation patterns. Participants read the sentences containing the target verbs along with the audio recordings of the sentences read by the first author, a native speaker of Mandarin Chinese (auto play). The participants were instructed to judge whether the sentences were grammatical via button press. They had 10 s for each sentence. The sentences were pseudo-randomized so that those from the same pair did not appear on the screen consecutively.

The task consisted of 66 pairs of sentences with 22 pairs for each verb type and 2 pairs for each of 11 separation patterns (3 verb types × 11 separation patterns × 2 pairs). Each pair of the sentences shared a target verb, a separation pattern and a sentence meaning, but differed grammaticality. In other words, the ungrammatical sentences intended to say something similar to their grammatical counterparts but used separation patterns

incorrectly. For example, in (13), both sentences mean "Xiaomei was sleeping" with the target VO verb 睡觉 (*shuì jiào*, sleep (V)-sleep (O), sleep) and the target insertion pattern, the aspect marker ZHE. The only difference is that ZHE was inserted within the verb in (13a) but followed the verb in (13b), which was ungrammatical.

| (13) | a. | 小 | 美 | 睡 | 着 | 觉。(grammatical) |
|---|---|---|---|---|---|---|
| | | *xiǎo* | *měi* | *shuì* | *zhe* | *jiào* |
| | | Xiaomei | | sleep (V) | ZHE | sleep (O) |
| | | 'Xiaomei was sleeping.' | | | | |
| | b. | 小 | 美 | 睡 | 觉 | 着。(ungrammatical) |
| | | *xiǎo* | *měi* | *shuì* | *jiào* | *zhe* |
| | | Xiaomei | | sleep (V) | -sleep (O) | ZHE |
| | | 'Xiaomei was sleeping.' | | | | |

In the example (14), the grammatical sentence contained a target VV verb 运动 (*yùn dòng*, move-move, 'exercise') and a non-separation pattern, i.e., a canonical subject–verb structure, while the ungrammatical sentence mistakenly contained a separation pattern that could only be used for VO verbs rather than VV verbs.

| (14) | a. | 小 | 美 | 在 | 操 | 场 | 上 | 运 | 动 | 着。(grammatical) |
|---|---|---|---|---|---|---|---|---|---|---|
| | | *xiǎo* | *měi* | *zài* | *cāo* | *chǎng* | *shàng* | *yùn* | *dòng* | *zhe* |
| | | Xiaomei | | on | the | playground | | move (V)-move (V) | | ZHE. |
| | | 'Xiaomei is exercising on the playground.' | | | | | | | | |
| | b. | 小 | 美 | 在 | 操 | 场 | 上 | 运 | 着 | 动。(ungrammatical) |
| | | *xiǎo* | *měi* | *zài* | *cāo* | *chǎng* | *shàng* | *yùn* | *zhe* | *dòng* |
| | | Xiaomei | | on the playground | | | | move (V) | -ZHE | -move(V) |
| | | 'Xiaomei is exercising on the playground.' | | | | | | | | |

5.2.5. Oral Translation Task

This task examined whether the participants could produce sentences with verb separation patterns accurately. Participants were instructed to say aloud the Chinese translations of English sentences with the target verbs and the separation patterns provided in the instruction for each trial in 15 s. Take 弹琴 (*tán qín*, to play-piano, 'to play piano') as an example. The participants read the English sentence "I am good at playing piano", the key verb "弹琴—tán qín—to play piano" and the instruction "please use proper adverb and/or classifier phrases in your translation". The target translation was 我弹琴弹得很好 (*wǒ tán qín tán de hěn hǎo*. I-play-piano-play-DE-very well. 'I am good at playing piano').

The task contained 66 sentences, i.e., 22 sentences per verb type and 2 sentences for each separation pattern (3 verb types × 11 patterns × 2 sentences). The same as in the grammaticality judgment task, the 11 patterns required at least one of the three types of verb compounds to separate. Among the 66 stimuli, 28 required the participants to use verb separation patterns in their translation. Among them, 18 sentences required the separation of VO verbs, 4 sentences required the separation of VV verbs, and 6 sentences required the separation of VR verbs. The trials were pseudo-randomized so that the sentences requiring the same separation patterns did not appear consecutively with each other.

The participants' translations were audio recorded and automatically uploaded to FindingFive Team (2019) servers. Two native speakers of Mandarin Chinese rated the produced sentences. The agreement rate on verb separation correctness was 99.9%. Specifically, a sentence was judged "correct" only if the instructed verb separation pattern was correctly used and "incorrect" if the pattern was incorrectly used (or omitted). We note that we rated the grammaticality of the produced sentences and found that 68.18% were fully grammatical. Within these, 60.20% of the sentences that required separation patterns were grammatical. Among these, 24.79% actually contained separation patterns. In other words, 75.21% of the produced sentences that required verb separation according to the instructions were grammatical, but they made use of alternative non-separation patterns rather than the target separation patterns.

*5.3. Procedure*

To eliminate carryover effects, three task orders were created for the main experiment, in which the three tasks were presented in different orders:

Order A: Speeded verb translation task → oral translation task → grammaticality judgment task

Order B: Oral translation task → grammaticality judgment task → speeded verb translation task

Order C: Grammaticality judgment task → speeded verb translation task → oral translation task

Participants were randomly but evenly allocated into the three orders of tasks. Participants first completed the LEXTALE_CH and LHQ3. At the beginning of the main experiment, a microphone test was performed to ensure that the participants' microphones were well connected with their laptops or desktops. The participants were instructed to read aloud and to audio record a sentence presented on their screens. After the microphone test, the participants started the main experiment. There was a short practice session containing one to three trials before each task and a brief break between the three tasks. Participants were required to complete the whole experiment in one session in five days. They would receive a reminder if they did not complete the experiment one day before their deadlines. Participants sent emails to the experimenter/the first author whenever they had questions or technical problems. To ensure that the participants were focused on the experiment, math questions were inserted as catch trials every 11 trials in the three main tasks. The main experiment was implemented on FindingFive Team (2019).

After the participants completed the experiment, they received an Amazon e-gift card as compensation. All tasks were approved by the authors' Institutional Review Board, and all participants gave consent before participating.

## 6. Results

Before the data analysis, two participants' data were completely removed because one only responded to 75% of the trials in the speeded verb translation task, while the other one had much lower accuracy than the chance level, 50% in the grammaticality judgment task. The remaining 28 participants' data were analyzed. The mean corrected accuracy of LEXTALE_CH scores was 15.52 (SD = 15.84). The mean LHQ3 aggregated score of global Chinese proficiency equaled to 0.61 out of 1 (SD = 0.22), which means that the L2-Chinese learners self-identified their Chinese proficiency as mildly above intermediate level.

For the decomposition and grammaticality judgment tasks, multiple linear regression models were built in the R environment (R Core Team 2021) in order to test the effects of verb type, Chinese proficiency, task order and the interaction between verb type and L2 proficiency. Task order was included in the regression models as an independent variable because it had been proved a potential confounding factor by extensive psycholinguistic research. Verb type and task order were dummy coded with VO verbs and Order A as reference levels, and then the regression models were releveled with VR verbs and Order B as reference levels.

*6.1. Speeded Verb Decomposition Task*

We collected 5925 responses in total in the task. Before the data analysis, an additional 3.93% of the data was removed due to internet instability at the participants' ends, or no responses within the time limit. Figure 1 shows that the mean accuracy of VV verbs (mean = 27.75%, 95%CI [23.47%, 32.02%]) was the lowest, whereas the accuracies of VR and VO verbs were higher (VR verbs: mean = 39.51%, 95%CI [34.58%, 44.45%]; VO verbs: mean = 46.66%, 95%CI [40.74%, 52.57%]). The mean accuracy of decomposition across all verb types was strikingly low (mean = 37.97%, 95%CI [34.68%, 41.26%]). Figure 2 shows that the mean accuracy of decomposing the meanings of the verbs increased linearly with the increase in L2 Chinese proficiency.

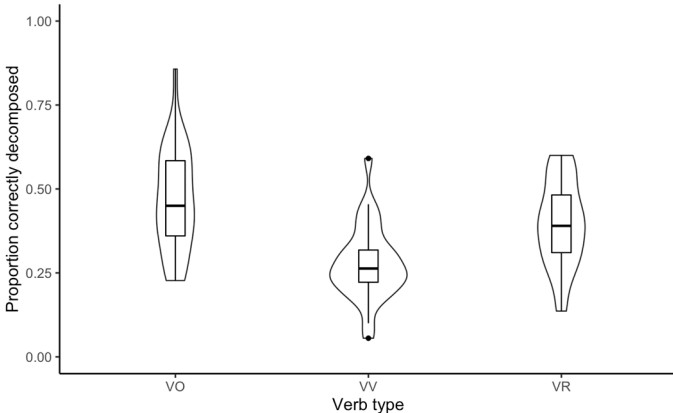

**Figure 1.** The mean accuracies of decomposing verbs across three verb types.

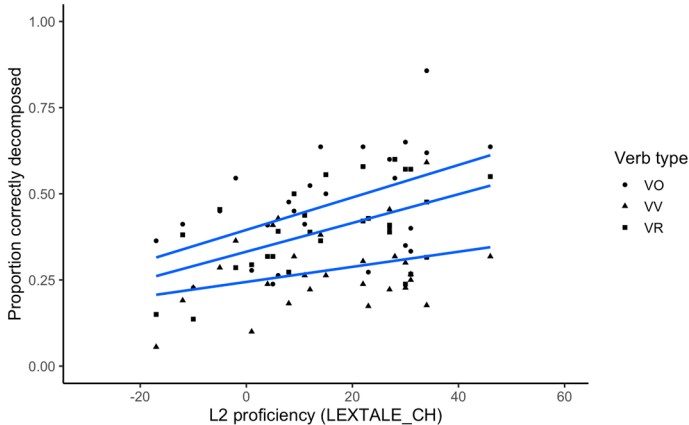

**Figure 2.** The mean accuracies of decomposing verbs across L2 proficiency.

The results of the multiple linear regression model ($F(7, 76) = 8.42$, $p < 0.0001$, *adjusted-$R^2$* = 38.48%) showed a main effect of verb type. The mean accuracy of decomposition for VV verbs was significantly lower than those for VO ($\beta = -0.15$, *SE* = 0.04, $t = -3.43$, $p = 0.001$) and marginally for VR verbs ($\beta = -0.09$, *SE* = 0.04, $t = -1.99$, $p = 0.04995$), whereas VO and VR verbs showed similar decomposition accuracies ($\beta = -0.06$, *SE* = 0.04, $t = -1.44$, $p = 0.16$). An effect of L2 proficiency was found ($\beta = 0.005$, *SE* = 0.001, $t = 3.28$, $p = 0.002$), indicating that as L2 proficiency increased, so did participants' ability to decompose the verbs accurately. Verb type showed a null interaction with L2 proficiency ($p$'s > 0.05). Lastly, task order showed null effects ($p$'s > 0.05), indicating that the participants from the three task orders had similar decomposition accuracies.

*6.2. Grammaticality Judgment Task*

We collected 3963 responses in total in the task. An additional 3.41% of the data was removed due to network errors, internet instability at the participants' ends, or no responses within the time limit. Figure 3 shows similar accuracies of processing the sentences with the three types of the verbs (VO verbs: mean = 61.31%, 95%CI [57.53%, 65.10%]; VV verbs: mean = 65.60%, 95%CI [62.44%, 68.76%]; VR verbs: mean = 62.80%, 95%CI [59.10%, 66.51%]). Figure 4 shows that proficiency had a linear effect on grammaticality judgment. The judgment accuracies of the participants with the lower L2 proficiency were around 50%, the chance level.

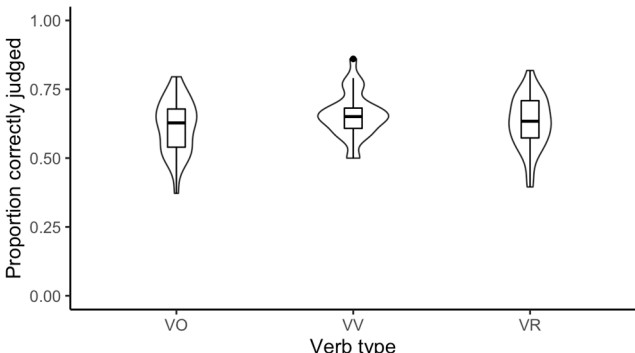

**Figure 3.** Grammaticality judgement accuracy across the three types of verbs.

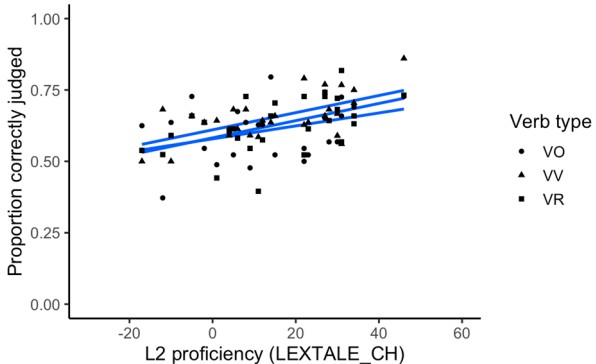

**Figure 4.** Grammaticality judgement accuracy along L2 proficiency.

The results of a multiple linear regression model ($F(7, 76) = 5.35$, $p < 0.0001$, *adjusted-$R^2$* = 26.83%) showed null effects of verb type (VR vs. VO verbs: $\beta = 0.004$, $SE = 0.03$, $t = 0.13$, $p = 0.90$; VV vs. VO verbs: $\beta = 0.03$, $SE = 0.03$, $t = 1.08$, $p = 0.28$; VV vs. VR verbs: $\beta = 0.03$, $SE = 0.03$, $t = 0.96$, $p = 0.34$). A significant effect of L2 proficiency was found ($\beta = 0.002$, $SE = 0.001$, $t = 2.50$, $p = 0.01$), indicating that as proficiency increased, so did grammaticality accuracy. The interaction between verb type and L2 proficiency was null ($p$'s > 0.05). Lastly, significant effects of task order were found. The mean accuracy of the participants from Order C was the lowest (Order B vs. Order A: $\beta = 0.01$, $SE = 0.02$, $t = 0.60$, $p = 0.55$; Order C vs. Order A: $\beta = -0.05$, $SE = 0.02$, $t = -2.11$, $p = 0.04$; Order C vs. Order B: $\beta = -0.06$, $SE = 0.02$, $t = -2.68$, $p = 0.009$). This may be caused by the fact that 11 participants came from Order A; 12 from Order B; but only 6 from Order C. The uneven distribution of the participants in the three orders is due to the removal of the participants who did not meet the inclusion criteria for the data analysis.

### 6.3. Oral Translation Task

In total, 1980 responses were collected. Figure 5 shows that the participants produced very few sentences with correct verb separation patterns. Four participants produced no grammatically correct sentences with separation patterns, only two participants produced 12 sentences correctly, and no participants produced all 28 sentences with verb separation correctly. In total, 12.70% (95%CI [9.24%, 16.16%]) of the sentences requiring VO verb separation were correct; for VV verbs, 31.25% (95%CI [20.17%, 42.33%]) were correct; and for VR verbs, 30.95% (95%CI [21.53%, 40.38%]) were correct.

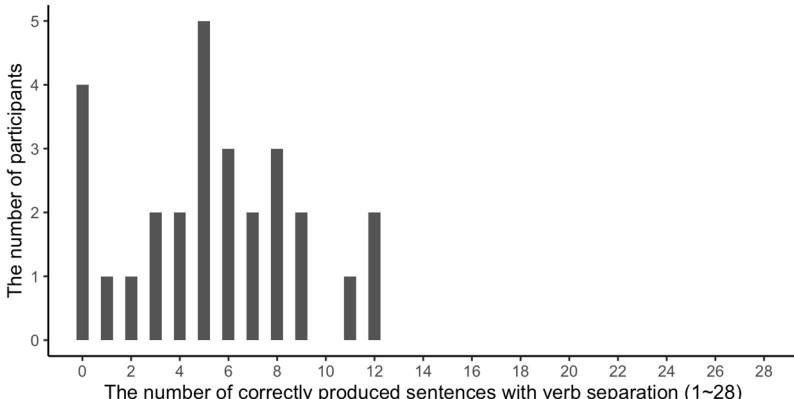

**Figure 5.** The number of the sentences with separated verbs correctly produced by the participants.

Figure 6 shows the production accuracy of verb separation increased linearly with the increase in L2 proficiency. A multiple linear regression model ($F(3, 24) = 2.73$, $p = 0.07$, *adjusted-R²* = 16.13%) showed an effect of L2 proficiency ($\beta = 0.004$, *SE* = 0.001, $t = 2.61$, $p = 0.02$). A null effect of task order ($p$'s > 0.05) was found. Different from the two tasks above, verb type as an independent variable was excluded from this regression model because the numbers of the trials across the three verb types varied markedly: VO verbs = 18, VV verbs = 4, VR verbs = 6.

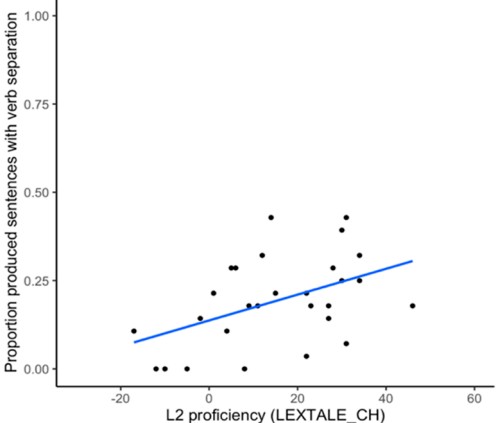

**Figure 6.** Rate of correctly produced sentences with verb separation across L2 proficiency.

## 7. Discussion

This study set out to answer three research questions related to adult L2 acquisition of Chinese verb separation. The first question asked what do L2 learners know about the decomposition of verb compounds? The 28 participants tested had significant difficulties in decomposing verbs. On average, learners were only able to decompose the meaning of 37% of the verbs in terms of understanding the bimorphemic verb and both constituent morphemes. Learners decomposed more VO and VR verbs than VV verbs. The results indicate that decomposing verbs, especially VV verbs, is a challenge for L2-Chinese learners.

The low level of ability to decompose verb compounds can be explained by the fact that L2-Chinese learners have not fully learned the meaning of the component morphemes. Whereas L1 speakers of Chinese can use their prior understanding of the component morphemes to process Chinese compounds (e.g., Tsang and Chen 2014; Wu et al. 2017), L2 learners of Chinese can only do this for the most transparent noun compounds, but not for less transparent compounds (Chen et al. 2020; Gao 2020). In theory, learners might be able to separate verb compounds without decomposing the meanings. However, if they understand that these compounds are composed of two separate parts, they will feel more comfortable about separating these parts. Thus, the ability to decompose these

compounds is an important first step for learning verb separation. For learning the overall word meaning, rote memorization or self-learning strategy are excellent strategies (e.g., Barcroft 2002; Van Hell and Mahn 1997; Wei 2015). Learners are good at memorizing multimorphemic words as units by rote. However, rote memorization naturally blocks the learning of compound decomposition.

The second question asked how do L2 learners process and produce the sentences containing verb separation patterns? Interestingly, the accurate decomposition of the verbs did not guarantee that the participants accurately processed the sentences containing VO and VR verbs or that they accurately produced them orally. The participants recognized similar percentages of sentences containing one of the three types of verbs. The grammaticality judgment accuracy results indicate that L2 learners have a fairly limited understanding of Chinese verb separation grammar. Most learners tested were only slightly above chance (average 63% accuracy). This indicates considerable room for improvement in both the teaching and learning of verb separation. The explicit instruction of L2 grammar (e.g., Tolentino and Tokowicz 2014) and potentially even prosody (e.g., Wiener et al. 2020) such as breaks before separated verbs may benefit learners' acquisition of these patterns.

Participants orally produced very few sentences with correct verb separation, and the sentences requiring VO verb separation showed the least production accuracy. One possibility is that VO verbs have the most complex separation patterns. Another possibility is that the participants we tested had not developed the fluency necessary to use the separation patterns and consequently avoided using them in the oral translation task. Regardless, the results indicate that spoken verb separation is a major problem for L2 learners. Limited evidence suggests oral production accuracy in L2 Chinese learners is typically equal to or slightly worse than auditory perception accuracy (Liu and Wiener 2021, 2022). This evidence, however, comes primarily from monomorphemic nouns. The present study presents initial evidence for the claim that production abilities lag far behind perception abilities (visual and/or auditory since both were presented in the tasks) in terms of L2 learning of morpho-syntactic constructs.

The third research question asked whether certain types of verbs would be more difficult than others. As noted above, decomposition was most difficult for VV verbs, and oral production was most difficult for VO verbs. Difficulties with VV verb decomposition may arise from a lack of L1 transfer. L2 learners can analogize VO verbs with verb phrases in European languages, but they cannot find such analogs for VV or VR verbs. Given this, decomposition accuracies of VV and VR verbs should also be lower than VO verbs. However, unlike VV verbs, the constituent morphemes of VR verbs are very productive (Zhang 2014). Learners may have already learned the morpheme meanings or even the decomposability of the VR verbs. As discussed above, the results of the oral translation task showed that VO separation was the most difficult to produce, in part because VO verbs had the most complex separation patterns.

Regardless of the tasks, we found robust effects of LEXTALE_CH score as a predictor of performance, indicating that as proficiency increased—here, in terms of a larger knowledge of word forms—learners' accuracy increased. One explanation for the results is that a larger vocabulary, i.e., greater awareness of different verb types, contributed, in part, to the increased processing and production accuracy (McDonald 2006). Importantly, however, even the most advanced participants tested in our sample were nowhere near ceiling, suggesting that our learners still had much room for improvement.

In general, the results show that verb separation patterns are extremely challenging for learners, even for the VV verbs with the least complex separation patterns, and even for advanced Chinese learners. This assessment of the level of learning for these patterns is further supported by comparing the participants' performance in the grammaticality judgment (processing) and the oral translation (production) tasks across the 11 separation patterns (Table 2). Reduplication and post-repetition with adverbs for frequency and duration had the lowest processing accuracy, while insertion with the aspect marker LE, operators and BU had the highest processing accuracies. Participants produced zero

sentences accurately with topicalization and insertion with LE and operators, but they produced sentences with pre-repetition with A-not-A and post-repetition with adverbs for frequency and duration most accurately. The results may indicate a modality mismatch: the patterns that were hard for processing were easy for production, and vice versa. The only exceptions were that A-not-A was easy for both processing and production, whereas reduplication was difficult for both processing and production. The results confirmed that the most complex separation patterns contributed to the lowest production accuracy of the VO verbs. They also revealed a wide range of the accuracies for each separation pattern.

**Table 2.** Summary of the mean accuracies with 95% confidence intervals in the brackets of processing and producing verb separation by separation pattern.

| Grammatical Constructs | Separation Patterns | Processing | Production |
|---|---|---|---|
| Insertion | ZHE | 0.60 [0.55, 0.64] | 0.16 [0.04, 0.28] |
| | LE | 0.71 [0.64, 0.77] | 0 [0, 0] |
| | GUO | 0.60 [0.55, 0.66] | 0.05 [−0.03, 0.13] |
| | operators | 0.71 [0.64, 0.77] | 0 [0, 0] |
| | DE | 0.59 [0.53, 0.64] | 0.25 [0.10, 0.40] |
| | BU | 0.75 [0.70, 0.81] | 0.13 [0.02, 0.23] |
| Post-repetition | Adverb (frequency and duration) | 0.55 [0.49, 0.61] | 0.30 [0.16, 0.45] |
| | Adverb (status) | 0.61 [0.56, 0.66] | 0.18 [0.08, 0.27] |
| Pre-repetition | A-not-A | 0.64 [0.58, 0.70] | 0.54 [0.41, 0.68] |
| Reduplication | Delimitative | 0.55 [0.49, 0.61] | 0.15 [0.08, 0.22] |
| Topicalization | Topic | 0.65 [0.58, 0.71] | 0 [0, 0] |

The learning difficulties found in this study support the analysis provided by the Unified Competition Model (MacWhinney 1987, 1992, 1997a, 2012, 2018, 2021). The fact that the participants relied so heavily on non-separation patterns as alternatives to the separation patterns in the production task demonstrates the existence of a competition between the two groups of the patterns, and that the non-separation patterns show a dominant position in L2 learners' knowledge. In addition, the finding that even advanced learners had difficulties shows that the complexity of these competitions and mappings leads to a slow and long learning process, and that learners are receiving insufficient input data or explicit instruction to accelerate this process.

This work provides the first in-depth exploration of L2 learners' knowledge on Chinese verb separation and contributes to the small but growing body of literature on adult L2 learning and recognition of semantically transparent and opaque compounds (e.g., Chen et al. 2020; Gao 2020; Wu 2011; Yi 2022; Zhang 2014). What remains to be seen, and the current focus of our follow-up study, is to what degree the explicit instruction of verb separation can benefit learners. For instance, does drawing the learner's attention to the morphosyntax involved in the separation patterns improve learning? If so, does this type of intervention have long lasting effects and transfer to untrained verbs?

This preliminary study is not without its limitations. First, participants were tested online. Unlike a controlled laboratory experiment, an internet-based study sacrifices experimental control for ease of recruitment and testing. We note that all participants reported passed the attention checks included in the study, though we cannot confirm how many wore headphones or performed the task in isolation without distraction. Second, we did find an effect of task order in the grammaticality judgment task. As discussed, this is most likely due to an imbalance of participants tested in each task order rather than a more serious design flaw given that we did not find task order effects in the other two tasks. Third,

an unavoidable confounding factor is that elementary learners might not have enough knowledge of the Chinese language to process and produce the sentences containing verb separation. The mean grammaticality judgment accuracy of the participants with lower LEXTALE_CH scores was around 50%. This may indicate that those elementary learners have not acquired verb separation, or that the task is too challenging for them.

To conclude, the study reveals that verb separation is very challenging for classroom learners of Chinese, particularly in terms of decomposition and production. These findings provide support for the Unified Competition Model and point to a need for pedagogical tools to assist learners in mastering Chinese verb separation.

**Author Contributions:** Conceptualization, Z.G., S.W. and B.M.; methodology, Z.G.; software, Z.G.; validation, Z.G., S.W. and B.M.; formal analysis, Z.G.; investigation, Z.G.; resources, Z.G., S.W. and B.M.; data curation, Z.G. and S.W.; writing—original draft preparation, Z.G.; writing—review and editing, Z.G., S.W. and B.M.; visualization, Z.G.; supervision, S.W. and B.M.; project administration, Z.G.; funding acquisition, B.M. All authors have read and agreed to the published version of the manuscript.

**Funding:** This research received no external funding.

**Institutional Review Board Statement:** The study was conducted according to the guidance of the Declaration of Helsinki and approved by the Carnegie Mellon University Institutional Review Board (IRB) (30 June 2020).

**Informed Consent Statement:** Informed consent was obtained from all subjects involved in the study.

**Data Availability Statement:** The stimuli for the experiment can be found at the OSF project page: https://osf.io/s3u2z/ (accessed on 21 August 2022).

**Acknowledgments:** The authors thank Ping Li for his valuable suggestions on this project. Yueming Yu, Gang Liu, Tianxue Yao, Xiaomeng Li, Ding Wang-Bramlett, Tianyu (Sophie) Qin, Shuai Li, Feng Xiao and Yi Xu helped recruit participants during the pandemic. The authors also thank Jiaqi (Angela) Chen for helping with rating the audio recordings, and Qiong Li and Joy Maa for proofreading the experiment materials.

**Conflicts of Interest:** The authors declare no conflict of interest.

## Note

1      The article uses upper letters for functional words and markers.

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
