# Peer review of "Acquisition of Chinese Verb Separation by Adult L2 Learners"

_languages, doi:10.3390/languages7030225_

Round 1
Reviewer 1 Report
Major comments
This paper used three tasks to investigate the learners' knowledge of verb separation in Chinese and their ability to apply the process of verb separation. The results showed that verb separation is quite challenging for learners, and even advanced learners were not at ceiling in applying the process. The results were explained using the Competition Model of L2 acquisition.
This study is interesting in that a process that is less represented in textbooks was investigated and the lack of oral input did prevent the learners from acquiring verb separation. However, doubts may arise as to the usage of some verb separation processes are not obligatory and thus even if the materials were designed to test the ability of verb separation, the subject (and even native speakers) can choose other equivalent structures. For example, "I am good at playing piano" can be translated as "我擅長彈琴" as well as "我彈琴彈得好". A suggestion is that more obligatory verb separation processes can be used in the oral translation task. For example, the process of pre-repetition for yes-no questions (p.2) may be more obligatory than other processes.
Another related issue is that although all 11 patterns of verb separation were tested in the experiment, a comprehensive analysis of the difference across patterns was missing. It is well understood that the number of tokens used may not enough for statistical analysis. But it would make more sense and more interesting to see which pattern (or process, five in total) is more challenging for learners. Summary statistics would suffice in such an anslysis. Would the subjects produce more verb separation patterns in more obligatory patterns, such as pre-repetition for yes-no questions?
A minor question is about statistic method. Why linear regression model was used instead of linear mixed-effect model? Without taking the random effects of subjects into consideration, the factor effects may not be gauged precisely.
Minor comments
- line 298, page 5, "exposure of natural speech" should be "exposure to natural speech".
- line 503, page 9, (also line 593, page 12) "A main effect of L2-Chinese proficiency". For numerical predictors, if there is a significant effect, the effect is linear. There is only a main effect of a factor and only when the factor is deviance coded when reporting the estimates. If a main effect is really needed, then model comparison would better make the point. But for the case at hand, it is not appropriate to say "a main effect of L2-Chinese proficiency".
- The same also applies to the report of the effect of verb type in line 520, page 10. As the factor is dummy coded (in line 471, page 8), there is only a simple effect when the estimate is significant.
Author Response
Dear reviewer,
Thank you for your comments. Please see the attachment for our response.

Reviewer 2 Report
The paper is quite difficult to understand in general. It is obvious that the author(s) are knowledgeable on the topics discussed, but the empirical facts are extremely difficult to follow - the authors perhaps take for granted that all readers are familiar with verb separation in Chinese. I strongly recommend rephrasing the first sections.
Author Response
Dear reviewer,
Thank you so much for your comments. Please see our response in the attached file.

Reviewer 3 Report
The topic is a meaningful one and some detailed analysis has been provided. The manuscript can be improved in the following aspects:
1) On p. 1, it writes “morphemes and adverbs such as ZHE0, LE0, GUO0, NE0”: which of the examples would be considered an adverb? Looks like none of them?
2) On p. 2, it says “verb-resultative (VR) verbs which are composed of a verb and a resultative preposition character”: Why would 开 in a VR such as 拉开” be considered a proposition?
3) On p. 2, it is mentioned that “depending on the morphological structure of the verb compound or the morphological or/and syntactic cues and the function of the target sentence”: How does morphological or/and syntactic cues play a role in your study design? There seems to be little discussion on this in your study.
4) On p. 2, “textbooks do not provide sufficient linguistic analysis for learners to fully understand all 14 patterns”: But based on some current (effective) language learning pedagogy, it is not a good practice to overload learners with so much information at one time.
5) On p. 2: it says: “the whole concept of verb separation may seem quite foreign to speakers of languages that make no use of this type of structure”: Although other languages may not have disyllabic verbs like in Chinese, their verb phrases will allow similar grammar functions.
6) On p. 4, not sure if 2a) and 2d) would necessarily have the same meaning. Some would also interpret the “了” in 2d) as “啦”. Something to consider.
7) On p. 7, the theoretical model, i.e., the Competition Model was not discussed in the literature review? Did any previous study also used this model? If not, can you further explain why you chose to use it?
8) About the participants, on p. 8, it is mentioned that participants had learned Chinese from one semester to multiple years. Can you report on how many are beginners, intermediate learners, etc., respectively, based on their curricular levels? A breakdown of participants’ levels based on the LEXTALE_CH test results will also be helpful. The information will be important to better validify and contextualize the findings.
9) About study design, since students with different proficiencies all complete the same tasks, beginning learners (e.g., those who had only studied Chinese for one semester) may likely have not learned most of the words in the tasks. For instance, the word “弹琴” used in the study would be challenging for beginners. Given the very short time given to complete each of the tasks, it’s difficult to know what participants were focusing on, perhaps recognizing characters, processing the audio, etc. Perhaps little attention may be devoted to the target separation task? The point is that, please provide a clearer justification on why the same/similar word lists and sentences were used for participants with dramatically different proficiencies rather than having them process words that match their language ability?
10) On p. 9, it says that “participants needed to indicate whether the translations of the verbs and the constituent characters were correct or not”: It looks like two mini-tasks are involved in this judgment task and only five seconds are given? Is it possible that a participant thinks that the translation of the verb is correct but the translation of the constituent characters is incorrect?
11) On p. 16, it is argued that “Even though the instruction for each trial provided the English translation of the target Chinese sentences, the meanings and the pronunciations of the key verbs and the target separation patterns, the participants lacked the knowledge to put these cues together to produce a grammatical Chinese sentence with the separation patterns”: Once again, if a learner has not learned the word yet, it would be a lot of information to process at a time. Given that only 15 seconds were given to each item in the production task, it would be hard to know what learners were focusing on during those 15 seconds.
12) In the Discussion section, L1 influence was not discussed, which was mentioned in the Competition Model earlier in the study design section. More depth would also be needed to better interpret the findings.
13) There are also some technical issues:
· Perhaps the tones can be directly marked on the pinyin instead of using numbers?
· On p. 1, “the separation processes generates …: generate?
· p. 8 “we predict that Chinese verb separation should be very challenging 301 for L2 Chinese learners with L1-European languages”: better wording?
Author Response
Dear reviewer,
Thank you so much for your work. Please find our response in the attached file.

Round 2
Reviewer 3 Report
Most of the comments have been addressed.
On p.2, I don’t think 低 in降低 is a preposition either. To give example, 在in放在 would be a preposition. Perhaps just say a resultative component?
Author Response
Dear reviewer,
Thank you for your suggestion. We have changed the term to "resultative component". In addition, we added a discussion on the issue of compound-phrase boundary from Line 215 to Line 238 so as to clarify the scope of the study and our definitions of compounds and verb separation.